# Relationship between Physical Activity and Emotional Regulation Strategies in Early Adulthood: Mediating Effects of Cortical Thickness

**DOI:** 10.3390/brainsci12091210

**Published:** 2022-09-08

**Authors:** Jingjing Wu, Lina Zhu, Xiaoxiao Dong, Zhiyuan Sun, Kelong Cai, Yifan Shi, Aiguo Chen

**Affiliations:** 1College of Physical Education, Yangzhou University, Yangzhou 225127, China; 2Institute of Sports, Exercise and Brain, Yangzhou University, Yangzhou 225127, China

**Keywords:** physical activity, emotion regulation strategy, cortical thickness, early adulthood, mediating effect

## Abstract

We investigated the relationship between physical activity (PA) and emotional regulation strategies among college students to establish the mediating role of cortical thickness. A total of 60 university students (18–20 years old) were enrolled in this study. The International Physical Activity Questionnaire (IPAQ-L) was used to estimate PA levels. Based on the International Physical Activity Working Group standards, PA levels were divided into low, medium, and high PA groups; emotional regulation strategies were determined by the Emotion Regulation Questionnaire (ERQ), including the Cognitive Reappraisal Scale (CR) and the Expressive Suppression Scale (ES). Structural magnetic resonance imaging (MRI) was used to measure cortical thickness. Differences in use of the ES strategy among high, medium, and low PA groups were not marked. However, compared to the low PA group, the CR strategy was frequently used in the high PA group, with a thicker right hemisphere rostral anterior cingulate cortex (rrACC). PA levels were positively correlated with thickness of the rrACC cortex (r = 0.398, *p* = 0.002 < 0.05) and CR strategy (r = 0.398, *p* = 0.002 < 0.05), and negatively correlated with the ES strategy (r = −0.348, *p* = 0.007 < 0.05). The rrACC cortical thickness played a partial mediating role in the relationship between PA and CR strategy, accounting for 33.1% of total effect values. These findings indicate that although the negative correlation between PA and ES was not significant, the positive correlation between PA with CR was significant, and rrACC thickness played a partial mediating role in the relationship between PA and CR, providing new evidence toward comprehensively revealing the relationship between PA, rrACC cortical thickness, and emotion regulation strategies.

## 1. Introduction

Emotion regulation and emotion-generation processes are simultaneous [1]. In various schools, the generally accepted definition of emotion regulation is the process during which individuals exert influence on emotion occurrence, experience, and expressions [2,3,4,5]. During this process, individuals exhibit different emotion regulation strategies to regulate emotional responses. Due to advances in positive psychology, most studies have focused on determining how individuals use emotion regulation strategies to cope with emotions [6,7,8,9,10]. Emotion regulation strategies are used almost every day [11], and different strategies can have different effects on subsequent emotions. Among them, cognitive reappraisal (CR) and expressive suppression (ES) are the most common and effective emotion regulation strategies [12]. The most widely used emotion regulation scale was compiled by Gross et al. at Stanford University and used to assess the frequency with which individuals habitually use CR and ES strategies [13]. The use of negative emotion ES for adjusting negative emotions may enhance and prolong physiological responses, leading to emotional regulation disorders [14]. Compared with ES, CR can better reduce emotional experience and reduce physiological responses and activation of the sympathetic nervous system, while with ES, negative emotional experience does not really disappear, but is backlogged. Evidence from neuropathology also further suggests that long-term use of ES strategies is more likely to produce mood regulation disorders. Conversely, CR can effectively ameliorate negative emotions [15]. The social competition atmosphere has become increasingly intense, and young people are facing tremendous pressures from society. Due to multiple areas of competition in various life spheres, including in academics and career choices, college students who are about to enter society are often plagued by negative emotions, such as anxiety, stress, and depression, among others, along with the influence of many factors during the coronavirus pandemic period. Some students have difficulties venting their negative emotions, resulting in mental health crises [16]. Therefore, there is a need to assess the factors that affect their mental health. It has been reported that the higher the frequency of college students using CR, the higher the level of their mental health [17].

Physical activity (PA) has been proven to be an important factor influencing individual use of emotion regulation strategies [18]. PA refers to a series of physical activities that consume energy through skeletal muscle contraction, and it is most often measured using questionnaires [19]. Regular PA can improve individuals’ mental health [20,21,22,23,24] and their abilities for emotion regulation [25]. Low PA levels are often associated with depression [26,27,28], emotional disorders and other problems [29,30,31]. Severe anxiety has been correlated with difficulties in using CR strategies, while more anxious people may not often engage in PA [32,33,34]. Moreover, PA can increase the use of CR [35,36]. Individuals with an active physical life tend to use more CR strategies to cope with stressful events [37,38]; that is, PA is highly correlated with CR strategies.

Advances in brain imaging technology have provided new research perspectives for comprehensively revealing differences in the brain structures of individuals habitually using emotion regulation strategies. Cortical thickness is an important indicator for characterizing structural plasticity of the brain, reflecting the distance between the outer surface of gray matter and inner surface of white matter. Studies have investigated the association between cortical thickness and differences in habitual use of cognitive reappraisal and expression inhibition [39,40,41]. Two strategies have been highly correlated with neural activities in the prefrontal cortex system, including the dorsolateral prefrontal cortex (DLPFC), dorsal anterior cingulate cortex (dACC)/paracingulate cortex, and the ventromedial prefrontal cortex (VMPFC) [42,43,44,45]. Cortical thinning of the left dorsolateral prefrontal cortex (dlPFC) and left ventrolateral prefrontal cortex (vlPFC) in adolescent females is significantly associated with more use of CR [46]. Moreover, significant correlation between ES and the superior frontal gyrus, including the medial prefrontal cortex, precuneus, and parahippocampal gyrus, has been reported [47]. These findings provide strong empirical evidence for the correlation between cortical thickness and emotional regulation strategies. However, there is still heterogeneity in the relationship between emotion regulation strategies and cortex in different brain regions among college students, which needs to be explored urgently.

Sport psychology studies have shown that PA is associated with better developmental advantages of brain structures [48,49]. Previous cross-sectional studies have assessed the relationship between PA and cortical thickness. For instance, it has been documented that children with higher PA levels have greater reductions in thickness of the prefrontal cortex and parahippocampal cortex [50]. Another study reported a dose–response relationship between cerebral cortex thickness and regular PA in leisure time among the elderly, and the gain begins from partaking in a small amount of PA during leisure time [51]. In adults, more PA at moderate to intense levels is related to a thicker cortex in the temporal pole of the left hemisphere and superior frontal gyrus, and these regions tend to decline with age [52]. Some longitudinal studies have also confirmed that intervention of active PA has positive effects on brain structure [53,54]. However, one study showed that there were no significant changes in fMRI activation in the anterior cingulate cortex (ACC) for the participants after 9 month PA intervention [55]. After a 5-year exercise intervention, the values of cortical thickness in the supervised exercise groups were not statistically different from those in the control group over time [56]. The controversy may be due to the particularity of the participants and the inconsistency of the measurement methods. College students are in early adulthood, and maturation trends of the cerebral gray matter cortex changes from cortical thickening in childhood to cortical thinning [57]. Since the cortical thickness can effectively predict emotional regulation strategy use in early adulthood, a thinner left dorsolateral prefrontal cortex (dlPFC) and left ventrolateral prefrontal cortex (vlPFC) thickness can predict more use of CR, and a thinner superior frontal gyrus was positively correlated with ES for females, but negatively correlated for males [45,46]. Whether cortical thickness plays a mediating role in the relationship between PA and emotional regulation ability in college students deserves further analysis.

Both CR and ES can activate brain regions that are closely related to emotion regulation, implying that there is a complex relationship between PA, cortical thickness, and emotion regulation strategies. The relationship between the three has not been conclusively determined. In addition, there is a lack of research on college students, and the neural mechanisms of changes in emotional regulation strategies induced by PA have not been consistently described. Based on accumulating evidence, this study aims to use psychometric methods and structural magnetic resonance imaging technology to explore the relationship between PA levels and emotion regulation strategies among college students, and then to analyze the mediating role of cortical thickness in this relationship to find new potential neural pathways.

## 2. Materials and Methods

### 2.1. Participants

A total of sixty 18–20-year-old college students (28 males and 32 females) were recruited from a university in Yangzhou City, Jiangsu Province, China. The following inclusion criteria were used: (1) 18–20 years of age; (2) right-handedness, and (3) normal vision without color blindness. On the other hand, the exclusion criteria included: (1) any mental and/or physical disorders that limited PA and/or the research results; (2) substance abuse or the taking of anything that affected the nervous system in the previous 24 hours (i.e., drug, nicotine, alcohol); (3) MRI contraindications (i.e., metallic implants, claustrophobia, pacemakers, or contrast allergy). All participants received a compensation for their participation in the study and agreed to sign an informed consent form after receiving a detailed explanation of the experimental procedure. All research procedures were in accordance with the latest version of the Declaration of Helsinki with ethical approval from the Ethics and Human Protection Committee of the Affiliated Hospital of Yangzhou University (2017-YKL045-01).

### 2.2. Physical Activity Measurement and Grouping

The International Physical Activity Questionnaire-Long (IPAQ-L) was used to assess PA [58]. The questionnaire has 26 questions grouped into 6 items, including life, transportation, and PA in leisure time. Based on survey data, results were classified and graded according to degree of PA for further statistical research and analysis. The English version of the questionnaire was studied in 12 countries, and the results showed that the questionnaire was reliable and valid. In order to promote this PA measurement tool among Chinese-speaking people, it was translated into Chinese, and the reliability and validity was recognized among Chinese college students [59]. The PA levels for college students were evaluated based on energy consumption using the IPAQ [60], and assigned to low, medium, and high groups. This grouping standard not only considered total PA levels, but also the frequency in a week and time of each day for PA. The IPAQ involves PA in various fields of daily life. Therefore, the grouping standard recommended by the IPAQ working group is superior.

### 2.3. Measurement of Emotion Regulation Strategies

The emotional regulation questionnaire, developed by Gross et al. [13], was designed to assess the levels of two emotional regulation strategies, namely CR and ES. The questionnaire includes 10 items, of which items 1, 3, 5, 7, 8, and 10 measure CR, while items 2, 4, 6, and 9 measure ES. The questionnaire on a 7-point Likert scale requires subjects to score each item from 1 (completely disagree) to 7 (completely agree) according to their own situation. Chinese scholar Wang Li et al. gave college students (between the ages of 16 and 25, with an average age of 20.59 years old) a Chinese version of the emotional regulation scale compiled by Gross. The Chinese version has a high degree of internal consistency and validity, reaching the standard of the field of psychology. For CR, internal consistency reliability was 0.85, while retest reliability was 0.82. For ES, internal consistency reliability was 0.77, while retest reliability was 0.79 [61]. The higher the score, the higher the frequency of using emotional regulation strategies.

### 2.4. Structural Magnetic Resonance Data Acquisition and Cortical Thickness Data Processing

Structural magnetic resonance imaging was completed at the Affiliated Hospital of Yangzhou University, and all imaging data for brain structures were acquired using the GE Discovery MR750W 3.0T magnetic resonance imaging system. Scanning parameters for T1-MPRAGE structure imaging included pulse repetition interval = 7.20 ms, echo time = 3.06 ms, thickness = 1.00 mm, flip angle = 12°, acquisition matrix = 256 × 256, and scanning field of view = 256 × 256 mm.

Cortical thickness data were processed as: based on MATLAB (2013) platform, the CAT12 toolbox (<Introduction for CAT12 Toolbox> http://www.neuro.uni-jena.de/cat/ (accessed on 10 March 2021)) that is based on SPM (<Introduction for SPM Software> https://www.fil.ion.ucl.ac.uk/spm/ (accessed on 10 March 2021)) was used to process T1-MPRAGE data to reorganize cortical surfaces of 3D-T1WI images. i. Registration and standardization: 3D-T1WI image space registration to the standard anatomical space of the Montreal Institute of Neurology. ii. Segmentation: the registered 3D-T1WI images were divided into gray matter, white matter, and cerebrospinal fluid, and intensities of segmented gray matter images corrected. iii. Automated reorganization of brain surface: including cortical surface registration, expanded surface subdivision of folded surfaces, and automatic correction of topological defects. iv. Image quality control: homogeneity of all images was checked. v. Spatial smoothing: the size of the smoothing kernel was set at 15 mm to improve signal-to-noise ratio of the image. vi. The cortical template was automatically marked based on the Desikan–Killiany–Tourville (DKT) cortical surface map and cortical thickness value extracted [62]. Area labeling of cortical surface refers to FreeSurfer’s cortical surface algorithm.

Brain areas with statistical inter-group differences in cortical thickness were selected as brain region of interest (ROI). The average cortical thickness data of the ROI were extracted from the clusters. Using total intracranial volume (TIV) and gender as covariates, group analysis for comparing the cortical thickness of PA groups was performed using one-way ANOVA to extract the significant brain regions between high, medium, and low PA groups and the thickness values in CAT12, with cluster level FWE correction with a threshold at *p* < 0.05. A Desikan-Killiany atlas or DK40 template was used.

### 2.5. Statistical Analysis

All statistical analyses were performed using the Statistical Package for the Social Sciences (SPSS; SPSS Inc., Chicago, IL, USA) version 25.0 for Windows. The statistical significance threshold was set at *p* < 0.05. Behavioral test values are reported as mean ± SD. The chi-square test (χ^2^ tests) and one-way ANOVA were used to compare differences in demographic variables, PA, and emotional regulation strategies between high, medium, and low PA groups. Mann–Whitney U t-tests were applied to detect differences in characteristics between male and female students, and one-way ANOVA was used to access the difference of extracted cortical thickness values of ROI in three groups. The significance level was adjusted using a Bonferroni correction to 0.05/3 = 0.017.

The partial correlation analysis was then used to test the relationship between PA and cortical thickness of relevant brain regions. The correlation between cortical thickness and scores on ES and CR scales was then analyzed. Partial correlation analysis was used to control age and gender, and to evaluate the correlation of PA, cortical thickness (brain region with significant differences between groups) and emotion regulation strategies. Pearson correlation coefficients (r values) were calculated and *p* < 0.05 was considered significant. Process plug-in Model4 in SPSS was used to test the mediating role of cortical thickness in the relationship between PA and emotion regulation strategies. The deviation-corrected percentile bootstrap method was used to estimate the 95% confidence interval of mediating effects by sampling 5000 bootstrap samples. If the confidence interval did not contain 0, it indicated that indirect effects of the mediating model were significant. *p* < 0.05 was the threshold for statistical significance.

## 3. Results

### 3.1. Demographics, PA, Emotional Regulation Strategies and Differences in Cortical Thickness among the Groups

Demographic characteristics, PA, and emotion regulation strategies for each group are shown in Table 1. Differences in the number of male and female individuals, age, height, weight, and BMI in each group, as well as healthy way of life were insignificant (*p* = 0.130, *p* = 0.733, *p* = 0.426, *p* = 0.418, *p* = 0.669, and *p* = 0.560, respectively). Males and females were significantly different in PA and healthy lifestyle (*p* = 0.017 < 0.05 and *p* = 0.026 < 0.05), whereas there were no significant differences in CR and ES (*p* = 0.235 and *p* = 0.691 respectively); the comparison graph is presented in Figure 1. To validate the accuracy of the grouping method, one-way ANOVA was performed as a confirmatory method. In contrast, there were significant differences in PA between the groups (*p* < 0.001). After a post-hoc analysis, the PA of the high group was significantly higher than those of the medium (*p* < 0.001) and low PA groups (*p* < 0.001). Moreover, differences in CR scores between the high and low PA groups were significant (*p* = 0.004). However, differences in CR scores between the medium and high PA groups, as well as between the medium and low PA groups were not significant (*p* = 0.099 and *p* = 0.319, respectively). Furthermore, differences in ES scores among the high, medium, and low PA groups were insignificant (*p* = 0.354). In summary, emotion regulation levels in the high PA group were better than in the low PA group, only on the CR.

Effects of PA on total cortical thickness were evaluated. Total intracranial volume (TIV) and gender were used as covariates in brain-wide correlations between cerebral cortical thickness and PA. Brain area with statistical inter-group differences in cortical thickness and the thickness values were extracted in CAT12. Then, one-way ANOVA was performed in SPSS to access the difference in cortical thickness in three groups. The results showed that the rACC cortical thickness was thicker in medium and high PA groups than in low PA group. Differences in rACC cortical thickness between the high and low PA groups were significant (*p* < 0.017), however, these differences were insignificant between medium and high PA groups (*p* = 0.189 > 0.017) as well as between medium and low PA groups (*p* = 0.028 > 0.017). In summary, college students with high PA had thicker rACC. The alpha level adjusted using the Bonferroni method was 0.017. The area of the brain that exhibited differences in rACC cortical thickness between the high and low PA groups and the thickness values are shown in Figure 2.

### 3.2. Correlations between PA with Cortical Thickness and Emotioaln Regulation Strategies

Partial correlation analysis with gender as controlled variable was used to evaluate dimensional correlations between PA with cortical thickness and emotional regulation strategies. Pearson correlation coefficients (r values) were calculated among PA, rrACC cortex thickness, and emotion regulation strategies. *p* < 0.05 was considered significant. It was found that: i. PA was positively correlated with rrACC cortex thickness (r = 0.398, *p* = 0.002 < 0.05) (Figure 3A); that is, the higher the PA level, the thicker the rrACC cortex. ii. rrACC cortical thickness was positively correlated with CR (r = 0.422, *p* < 0.001), but not significantly correlated with ES (r = 0.010, *p* = 0.938 > 0.05) (Figure 3B), implying that the thicker the rrACC, the higher the CR score, i. e., the more efficient the use of emotional regulation strategies. iii. PA was positively correlated with CR scores (r = 0.398, *p* = 0.002 < 0.05) (Figure 3C) and negatively correlated with ES scores (r = −0.348, *p* = 0.007 < 0.05) (Figure 3D).

### 3.3. Mediating Roles of Cortical Thickness in the Relationship between PA and CR Strategies

There were significant correlations between PA, rrACC cortex thickness, and CR strategy. PA was a good predictor of CR (β = 0.418, t = 3.51, *p* = 0.001 < 0.05). To further test our hypothesis, SPSS Model 4 (simple mediation model) in Hayes [63] was used to evaluate the mediating effects of rrACC cortex thickness in the relationship between PA and CR strategy. Inference of indirect (mediated) effects was subsequently assessed using bootstrap confidence intervals (CI). Significance of indirect effects was assumed if the 95% confidence interval (95%-CI) did not include a zero. The number of bootstrap samples was set to *n* = 5000. The model is shown in Figure 4.

After adding the mediation variable, predictive effects of PA on CR were still significant (β = 0.280, t = 2.20, *p* = 0.032 < 0.05). Positive prediction of PA on rrACC cortical thickness was significant (β = 0.439, t = 3.72, *p* < 0.001). The positive predictive effect of rrACC cortex thickness on CR was also significant (β = 0.315, t = 2.479, *p* = 0.016 < 0.05).

The upper and lower limits to bootstrap 95% confidence intervals of the direct effects of PA on CR and mediating effects of rrACC cortex thickness did not contain 0, implying that PA can predict CR directly or through the mediating effects of rrACC cortex thickness. Direct effects (0.239) and mediating effects (0.118) accounted for 66.9% and 33.1% of total effects (0.357), respectively.

## 4. Discussion

In this study, integration of behavioral and imaging techniques revealed that PA is closely associated with emotional regulation strategies among college students. College students with higher PA have a higher prevalence of using CR, and better development and maturity of rrACC cortex. Thickness of rrACC cortex plays a mediating role in the relationship between PA and CR. For the first time, we used the mediating model to explore the mediating role of rrACC cortical thickness in the relationship between PA and CR, and found potential neural pathways involved in the association, which provides new evidence toward comprehensively revealing the relationship between PA, rrACC cortical thickness, and emotional regulation strategies.

There were gender differences in the PA and healthy lifestyle scores. It may have been that in daily life, the female students were more concerned about diet and nutrition than male students, while males were more willing to engage in PA than females [64]. Gender differences between CR and ES were not significant. However, previous studies have shown that emotional regulation is influenced by gender. In particular, males are more inclined to use CR, which may explain why females are prone to developing depression [65]. The absence of significant differences in our study was probably because of our small sample size, but our research aimed to explore the relationship between PA and emotional regulation strategies, with students grouped by PA level. In future, studies on a larger sample may show gender differences. We used partial correlation analysis with gender as a controlled variable to assess the correlations between PA and emotional regulation strategies. The result was consistent with existing research evidence. The meta-analysis results of Hyde et al. showed that children aged 3–9, adolescents aged 10–17, young adults aged 18–25, adults aged 26–65, and the elderly over 65 years old had more positive emotions when they experienced regular PA [66]. The research proves that higher levels of PA are associated with lower Stroop interference (indicating greater domain-general cognitive control, such as inhibitory control) and enhanced CR success. Subjects may have better cognition function, which leads to better regulation of emotions [37]. Studies involving healthy adults in Europe and the United States showed that when gender, age, and other variables are controlled, individuals undertaking active PA have positive emotions approximately 20% more frequently than those without PA [67]. Based on the analysis, it can be concluded that PA has a close relationship with CR in early adulthood. Moreover, there are differences in rrACC cortical thickness among studied groups. A study in young people confirms that more medium-to-high-intensity PA per day is positively correlated with greater rostral anterior cingulate cortex (rACC) thickness in the left and right brain hemispheres [68], which is consistent with our study. Other brain areas where cortical thickness is strongly associated with PA have been discovered. Older adults who habitually engage in leisure-time PA have been shown to have a thicker temporal pole and superior frontal gyrus cortical thickness of the left hemisphere than older adults who undertake no leisure time PA. Correspondingly, neurodegeneration in adults who perform non-leisure time PA begins about 3 to 4 years earlier than in adults performing frequent leisure-time PA [51]. However, these regions did not yield positive results in our study, which may be due to differences in the study subject population. High levels of PA have also been associated with higher levels of neurotrophic factors, such as brain-derived neurotrophic factors [69], synaptic plasticity [70], and increased cerebral blood flow [71]. PA may also help maintain brain functions and resilience [72,73]. A study involving older adults predisposed to Alzheimer’s disease showed that higher levels of PA is primarily associated with a thicker fusiform gyrus and temporal cortex, particularly in participants who engage in PA per week, with at least 1500 met-min/week of high-intensity PA [74]. It is worth noting that previous research consists mostly of correlation analysis. Our findings are consistent with the theory in the field of motor-cognitive neuroscience that PA is closely associated with brain structure, and they complement the correlation analysis research by exploring differences in cerebral cortical development between high, medium, and low PA groups in early adulthood, providing a new perspective for in-depth analysis.

Furthermore, we found that college students with high PA levels frequently used the CR emotional regulation strategy, which is often associated with a thicker rrACC cortex. This is consistent with previous studies, where rACC, a region of the prefrontal cortex that performs an important role in regulating mood and motivation, is responsible for mobilizing and reinforcing optimism in the brain, and plays an important role in neural networks involved in emotional processing and decision-making [75]. Kanske P et al. measured neural responses to emotional images in relieved patients with previous episodes of major depressive disorder using functional magnetic resonance imaging. Patients with depression used the reassessment strategy when viewing pictures, which increased the rrACC cortex activities [76]. However, our results did not show that rrACC thickness among college students was associated with ES, which may be due to the fact that the participants were in early adulthood and the direct correspondence between their rrACC thickness and ES was low. Based on this, we postulated that high PA levels among college students, accompanied by a thicker rrACC cortex, were associated with a frequent use of the CR strategy.

The rrACC cortical thickness was a mediator, and the mediation model was complete. This suggests that PA can predict the use of CR directly or through the mediating effects of rrACC cortical thickness. In this study, results were bootstrap-corrected in the mediation test, the statistical methods were rigorous, and findings were reliable. The indirect effects of rrACC thickness in the relationship between PA and CR were found to be reasonable. Cortical thickness is one aspect of brain plasticity. We established that the influence of PA on brain plasticity is also reflected in brain morphology and function. Children with higher PA have an increased volume of gray matter in the frontal, temporal, and occipital lobes [37]. Flöel et al. [69], using voxel-based morphology and functional magnetic resonance imaging techniques, found that high PA levels are associated with increased volume of the prefrontal cortex and cingulate cortex. Second, rrACC thickness plays a mediating role in the relationship between PA and CR strategy, but processing of emotional regulation strategies relies on synergistic effects of multiple brain regions. Variations in gray matter volume in the insula, dACC, ventral medial prefrontal cortex (vmPFC), and right dorsal medial prefrontal cortex (dmPFC) may also be the basis for individual differences in using CR and ES [39,44]. Therefore, rrACC cortex thickness is only one area in the process of emotion regulation.

The mediation model was used in our study for the first time to investigate the mediating roles of cortical thickness in the relationship between PA and the use of emotional regulation strategies, and the potential neural pathways of this relationship were found. These findings reveal the relationships among the three variables, and the mediating roles of other relevant brain indicators in the relationship between PA and emotional regulation strategies should be considered in future. This study has some limitations. First, the sample size is small (*n* = 60), and during data collection, there were certain restrictions on time, personnel, site, and equipment, However, a previous study with a small sample size (*n* = 31) also obtained reliable evidence [77], and we will consider validating our results in a large sample size in further study. Second, we only used the survey method; therefore, more supporting methods, such as the pedometer combined questionnaire method, acceleration sensor combined with HR monitoring, etc., should be added later to prove the validity of the behavioral data. Although the effect of PA on the psychology of healthy adults is still controversial because of the matured functional and structural development of the human brain in early adulthood [78], we want to provide theoretical support for the hypothesis that sports can promote the physical and mental health of healthy adults, starting with college student populations. Limitations aside, even if changes in cortical thickness are a mechanism underlying some of the emotional regulation ability-enhancing effects of PA, it is still unlikely that the pathways underlying such changes are consistent across age groups. Future studies need to consider the mechanisms of PA at multiple levels, and consider mechanisms that might differ depending on the population under investigation, the brain region of focus, or the parameters of exercise being used, and it is possible to combine more psychological tests at the selection stage to determine the personality types, mood status, etc. of the participants, so that the characteristics of the group are more specific.

## 5. Conclusions

In summary, the negative correlation between PA and ES in college students was not significant, but positive correlations between PA and CR were significant among college students, and the rrACC cortical thickness played a partially mediating role in the relationship. These findings suggest that even in younger adults, PA can predict neurological differences leading to structural brain changes, such as greater cortical thickness in the rrACC, which may in turn forecast more active and efficient use of emotional regulation strategies. Our findings revealed the relationship between PA and emotional regulation strategies among college students and the potential neural mechanisms, suggesting that universities can carry out targeted interventions on students to improve their PA levels and improve or slow down changes in neurodevelopmental structures induced by insufficient PA, since universities are ideal settings for implementing health promotion programs. Therefore, planning and implementing programs to motivate students to be more responsible for their own health and to engage in more PA are of paramount importance. This may help them lay a foundation for better regulation of emotions when they encounter negative emotions, such as stress, in society.

## Figures and Tables

**Figure 1 brainsci-12-01210-f001:**
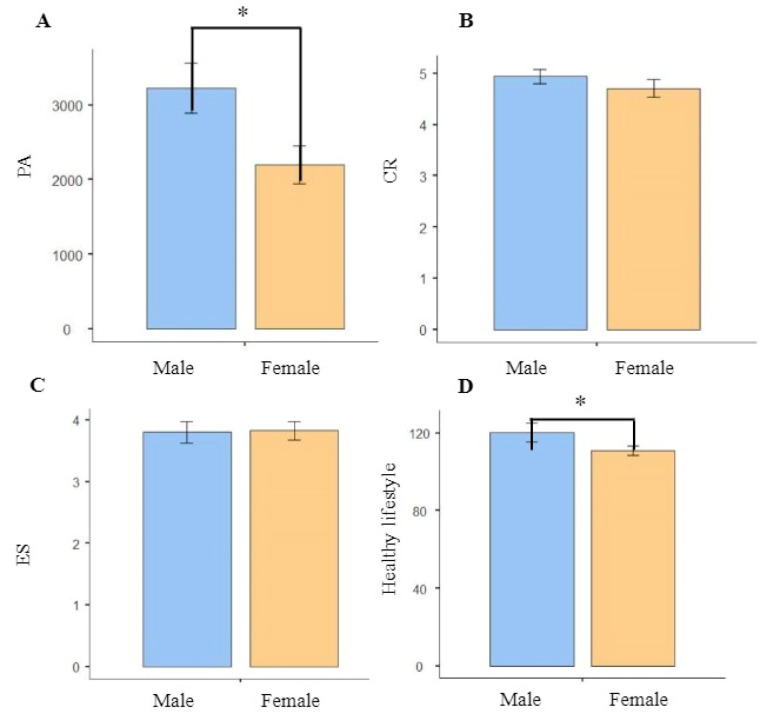
(**A**) Comparison graph of male and female in PA. (**B**) Comparison graph of male and female in CR. (**C**) Comparison graph of male and female in ES. (**D**) Comparison graph of male and female in healthy lifestyle. Abbreviations: PA, physical activity; CR, cognitive reappraisal; ES, expressive suppression. Note: * Denotes significant differences at *p* < 0.05.

**Figure 2 brainsci-12-01210-f002:**
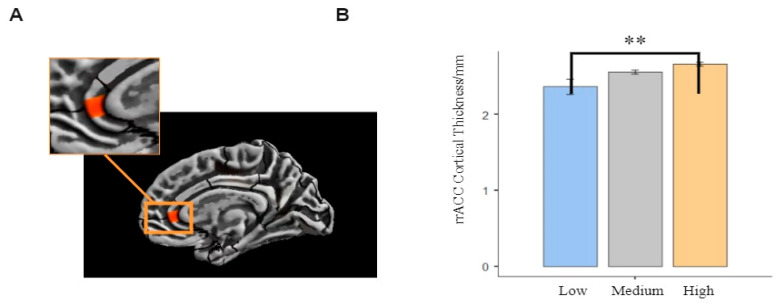
(**A**) Distinct regions that exhibited differences in cortical thickness (rostral anterior cingulate of right region) between high and low PA groups. (**B**) Surface-based morphometry results of cortical thickness in the three groups. Surface-based morphometry results after multiple comparisons correction. Note: ** Denotes significant differences at *p* < 0.01.

**Figure 3 brainsci-12-01210-f003:**
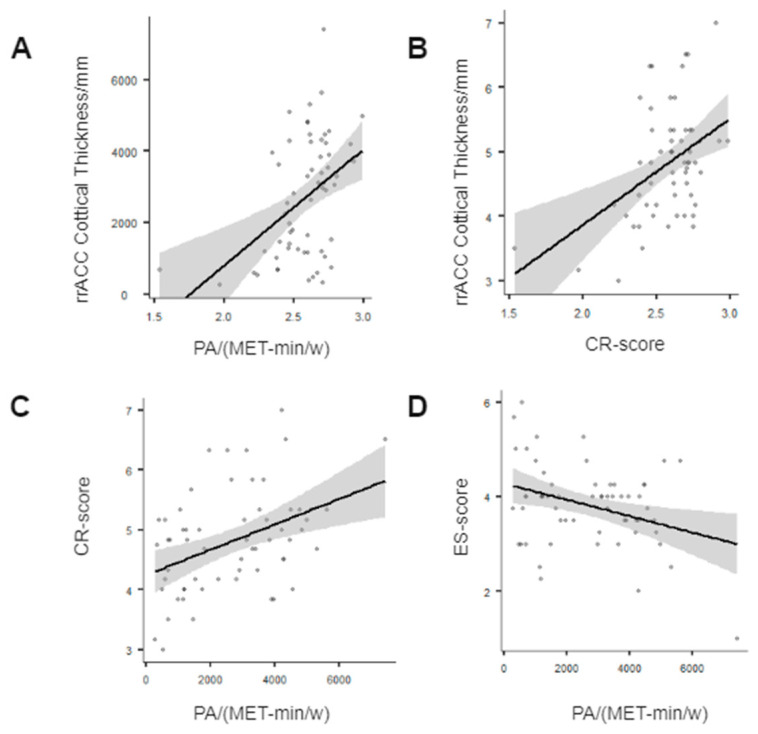
(**A**) PA levels were positively correlated with rrACC; (**B**) rrACC was positively correlated with CR; (**C**) PA levels were positively correlated with CR; (**D**) PA levels were negatively correlated with ES. Abbreviations: PA, physical activity; rACC, rostral anterior cingulate cortex; CR, cognitive reappraisal; ES, expressive suppression.

**Figure 4 brainsci-12-01210-f004:**
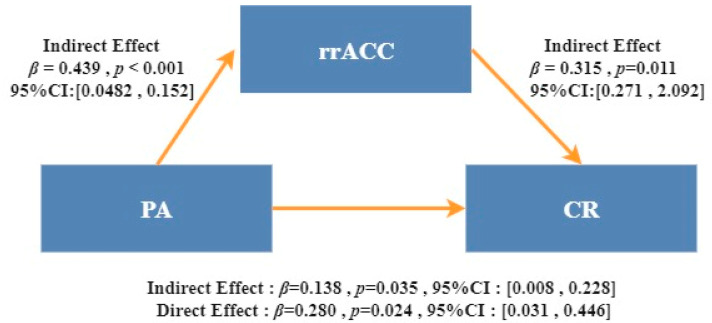
Mediating effects of rostral anterior cingulate cortical thickness (right region) on the relationship between physical activity and cognitive reappraisal. Abbreviations: PA, physical activity; rACC, rostral anterior cingulate cortex; CR, cognitive reappraisal; ES, expressive suppression. The values are standardized coefficients.

**Table 1 brainsci-12-01210-t001:** Participant demographics, physical activity, and emotional regulation strategies.

Variable	Low PA Group	Medium PA Group	High PA Group	*p* Values
Number	12	19	29	
Gender (Male/Female)	3/9	8/11	17/12	0.130
Age	19.50 ± 0.522	19.47 ± 0.513	19.59 ± 0.501	0.733
PA	592.96 ± 241.76 ***	1766.91 ± 703.08 ***	4142.26 ± 998.30 ***	*p <* 0.001
CR	4.24 ± 0.74 **	4.67 ± 0.81 **	5.15 ± 0.79 **	0.004
ES	4.18 ± 1.02	3.79 ± 0.74	3.68 ± 0.86	0.354
Height	167 ± 6.62	167 ± 8.67	170 ± 8.50	0.426
Weight	59.5 ± 9.55	61.1 ± 11.90	63.7 ± 9.68	0.418
BMI	21.3 ± 2.66	21.9 ± 3.00	22.2 ± 2.58	0.669
Healthy lifestyle score	111 ± 18.53	114 ± 18.5	119 ± 22.6	0.460

Note: ** Denotes significant differences at *p* < 0.01. *** Denotes significant differences at *p* < 0.001.

## Data Availability

The data presented in this study are available on request from the corresponding author. The data are not publicly available due to privacy of participants.

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
