# Peer review of "Relationship between Physical Activity and Emotional Regulation Strategies in Early Adulthood: Mediating Effects of Cortical Thickness"

_brainsci, 2022, doi:10.3390/brainsci12091210_

Round 1

Reviewer 1 Report

I appreciate you work and the opportunity to help as a reviewer. I made a number of comments. They vary in importance. Again, thank you for your efforts. I hope my comments help.

Abstract, please provide a few statistics whether they are p < .xx or some helpful effect size values. Then readers who browse abstracts will gain more information. For instance, you can provide the correlation values. Those are effect size values, as I know you know.

Introduction
  In text references - I think you need a space each time. last word[ref#] to last word [ref#].   Line 48 - Conversely, CR can effectively... Can you tell us how it does this? A bit more information is required to help the reader.   Line 86-87 - Please provide the references for your first sentence at the end.   Line 86-104 - For a balanced paragraph, are there any studies showing the relation does not exist?   Line 106 - Please explain in a few sentences why the relationship is complex. It seemed very positive and now it is complex.   Line 127 - Should it not be The International Physical...   Then line 134 given you have used IPAQ-L many times, you should keep using it.   Line 44 - you wrote Gross et al. and then line 140 you wrote Gross and John.   Results - just my opinion. If you write 0.13 and 0.733 you should write 0.130.   Line 241 - I do not understand how or why age could matter. Please explain how it could be an important covariate or remove and provide the new statistics. Your stated age range is 18-20.   Table 2 and Figure 3 - are they not the same statistics? There is not a need to repeat them in both the figure and table. The same for Table 3 with the same information as you wrote in the figure.   Can you provide a clear study limitations and future directions section? Those seem needed.

Reviewer 2 Report

The manuscript studies the relationship between physical activity (PA) and emotion regulation strategies in a sample of 60 university students to establish the mediating role of cortical thickness. The sample is low for a cross sectional study but recruitment criteria are correctly described. The International Physical Activity Questionnaire (IPAQ-L) was used to estimate PA levels of physical activity, divided into 3 groups, low, medium and high PA. The emotion regulation strategy was measured with the emotion regulation questionnaire (ERQ), which includes the Cognitive Reappraisal Scale (CR) and the Expressive Suppression Scale (ES).The study is cross-sectional and involved the use of magnetic resonance imaging (MRI) to measure cortical thickness. Differences in the use of the ES strategy among the high, medium and low BP groups were not detected. However, compared to the low BP group, the CR strategy was frequently used in the high BP group, with anterior rostral cingulate cortex of the thicker right hemisphere (rrACC). PA were positively correlated with rrACC cortex thickness and CR strategy and correlated negatively with ES strategy. Cortical thickness rrACC played a partial mediating role in the relationship between PA and the CR strategy, which represent 33.1% of the total effect values. These results indicate that although the negative correlation between PA and ES was not significant, the positive correlation between PA with CR was significant and the thickness of rrACC played a partial mediating role in the relationship between PA and CR, providing new evidence to fully reveal the relationship between PA, rrACC cortical thickness and emotion regulation strategy. The applied statistics are complete and compensate for the limitation of the sample size. The resampling method with the bootstrap method was well used. The results are statistically significant and the discussion also details study limitations.

Reviewer 3 Report

The work is undoubtedly interesting and relevant. In the study, the authors used a highly informative objective research method - cortical thickness – and compared it with data from several questionnaire concerning the regulation of emotions and physical activity. 

At the same time, the work requires undoubted refinement in terms of the description and detail of the methodological component, and the authors should also give satisfactory answers to the following questions in the text of the article:

1. In the opinion of the reviewer, a more thorough study of the division into "working" groups of students is required, since it was initially stated that the strategies for regulating emotions are very diverse and have an individual aspect.

2. The small sample size is doubtful (they say this, but doubts do not go away from this) - the results may be different with a larger volume, and gender differences may well manifest themselves, which the authors disown in the text. 

3. The age of the subjects is such that it does not seem to be equivalent to argue for the "insignificance of the influence of gender" of research in the USA of people with an already mastered place of work, life, etc. It should be understood that students are a special social group and they should be approached more carefully. It is not entirely clear at what period of study the students were examined – since this moment is of particular importance for them and in general the description of the observation group in the presented form cannot suit an attentive reader due to the paucity of objective information about the surveyed contingent – what kind of course it was, what kind of specialization in studies and other things need to be given in the description..

4. The "advantage" of respondents with a high level is insufficiently justified - how does it manifest itself in real life, outside of numbers? Are their academic performance higher? Are they calmer? Are you less prone to depression? There is not enough block that, after identifying high – medium - low indicators, would show something about them to convince that this is really the case.

5. The direction looks promising, but the execution is crude, we need to look for a more valid argument, it is possible to connect more psychological tests at the selection stage to determine personality types, student response models, so that the studied groups are more specific.

6. When the authors talk about physical performance, they use only the survey method – therefore, a section is required that convincingly proves that the survey method corresponds to real indicators of physical activity.

7. All the questionnaires used by the authors were designed for an English–speaking audience - it is unclear whether the adapted version was used for the surveyed Chinese students (and whether there is one) or whether the students were sufficiently proficient in English - at the same time, the level of proficiency in the original language of the questionnaires is not indicated.

Thus, the work requires finalizing the description of the methodological part of the study, as well as considering the doubts expressed by the reviewer in the text of the article.

Round 2

Reviewer 3 Report

The authors literally answered all the comments of the reviewer point by point, and the work looks more reasonable. At the same time, it seems that the very structure of the presentation of information should be revised. First of all, it is necessary to justify the absence of differences by gender - to give a comparison graph - to conclude that there are no differences, so the group can be formed mixed, and not to justify every time that "we did not see differences" after which it should be indicated that in this study we intend to show what we got, although in future studies on a larger sample may show gender differences. It is probably necessary to emphasize the practical importance of working in detention - which will give the same students their high level of physical activity, including right now, in their learning environment. Or will it only ever matter there? It is clear that students are, in a certain sense, the simplest object of research, but in the long run, what? Let them either justify that it is students with their unique position who are ideally suited to solve some research goals.
